# Ectopic Expression of Sugarcane *ScAMT1.1* Has the Potential to Improve Ammonium Assimilation and Grain Yield in Transgenic Rice under Low Nitrogen Stress

**DOI:** 10.3390/ijms24021595

**Published:** 2023-01-13

**Authors:** Shiwu Gao, Yingying Yang, Jinlong Guo, Xu Zhang, Minxie Feng, Yachun Su, Youxiong Que, Liping Xu

**Affiliations:** 1Key Laboratory of Sugarcane Biology and Genetic Breeding, Ministry of Agriculture and Rural Affairs, National Engineering Research Center for Sugarcane, Fujian Agriculture and Forestry University, Fuzhou 350002, China

**Keywords:** low nitrogen, sugarcane, ammonium transporter, rice, transgenic, nitrogen assimilation

## Abstract

In China, nitrogen (N) fertilizer is excessively used in sugarcane planting areas, while the nitrogen use efficiency (NUE) of sugarcane is relatively low. Mining and identifying the key genes in response to low N stress in sugarcane can provide useful gene elements and a theoretical basis for developing sugarcane varieties with high NUE. In our study, RNA-Seq combined with qRT-PCR analysis revealed that the *ScAMT1.1* gene responded positively to low N stress, resulting in the stronger low N tolerance and high NUE ability of sugarcane cultivar ROC22. Then, *ScAMT1.1* was cloned from sugarcane. The full-length cDNA of the *ScAMT1.1* gene is 1868 bp, containing a 1491 bp open reading frame (ORF), and encoding 496 amino acids. ScAMT1.1 belongs to the AMT superfamily and shares 91.57% homologies with AMT1.1 from *Oryza sativa*. Furthermore, it was stably overexpressed in rice (*O. sativa*). Under low N treatment, the plant height and the fresh weight of the *ScAMT1.1*-overexpressed transgenic rice were 36.48% and 51.55% higher than that of the wild-type, respectively. Both the activity of ammonium assimilation key enzymes GS and GDH, and the expression level of ammonium assimilation key genes, including *GS1.1*, *GS1.2*, *GDH*, *Fd-GOGAT*, and *NADH-GOGAT2* in the transgenic plants, were significantly higher than that of the wild-type. The grain number and grain yield per plant in the transgenic rice were 6.44% and 9.52% higher than that of the wild-type in the pot experiments, respectively. Taken together, the sugarcane *ScAMT1.1* gene has the potential to improve ammonium assimilation ability and the yield of transgenic rice under low N fertilizer conditions. This study provided an important functional gene for improving sugarcane varieties with high NUE.

## 1. Introduction

Nitrogen (N) is one of the essential elements for crop growth and development [1]. N supply is a critical limiting factor of high yield in crop production, resulting in much chemical N fertilizer used in crop production worldwide. Sugarcane (*Saccharum* spp. hybrids) is the most important sugar crop. As a tall perennial grass, sugarcane requires a large quantity of N for normal growth, and excessive chemical N fertilizer has been applied to increase sugarcane yield, leading to the low nitrogen use efficiency (NUE), with only 20–40% absorbed of the N application amount in sugarcane production [2,3,4], especially in China and India, two of the top three sugarcane producing countries [5]. This not only causes soil acidification and compaction but also reduces the sugar content of the sugarcane. The breeding of high NUE varieties is an effective way to reduce N fertilizer consumption [6]. Identification of the key genes and studying the physiological and molecular mechanism regulating N metabolism in the sugarcane response to low N stress can provide key gene elements for breeding sugarcane varieties with high NUE.

Ammonium (NH_4_^+^) and nitrate (NO_3_^−^) are the main inorganic N sources absorbed by crops. The roots of most crops mainly absorb NO_3_^−^. However, the energy required for NH_4_^+^ uptake by roots is less than that for NO_3_^−^ uptake [7,8]. In the case of N deficiency, the roots of sugarcane and rice preferentially absorbed NH_4_^+^ [5,9]. In order to adapt to soil environments with different N concentrations, plants have evolved two types of N uptake and transport systems: low- and high-affinity transport systems. The former plays a major role in N uptake under high N concentration and the latter plays a major role in N uptake under low N concentration [10,11]. The uptake and transport of NH_4_^+^ by plant roots are mediated by ammonium transporters (AMT) [10,12]. The AMT family is mainly divided into the AMT1 (AMT1 cluster) and AMT2 (AMT2/3/4 cluster) subfamilies [7,13]. AMT1-type ammonium transporters mainly mediate high-affinity ammonium uptake in plant roots, which has been well demonstrated in various species by previous studies [1,9]. Most AMT2 subfamily members belong to the low-affinity transport system [14]. In *Arabidopsis*, in addition to contributing to root uptake in the low-affinity transport system, the *AtAMT2.1* gene plays a role in the root-to-shoot translocation of ammonium [15]. Due to the low homology with the AMT1 subfamily, the AMT2 subfamily may have different biological functions from that of the AMT1 [7].

Within the AMT1 family, the *AMT1.1* gene has been shown to respond positively to low N stress and play an important role in the process of NH_4_^+^ uptake in crop roots [9,16]. In barley, there was a positive correlation between *AMT* gene expression (*AMT1.1* was expressed 2.8 times more than *AMT2.1*) and the N content in the leaves at anthesis and the 1000-kernel weight [17]. The *AtAMT1.1* gene of *Arabidopsis* was strongly expressed in the roots under the N-deficiency stress. However, it was strongly expressed in the leaves when the stress disappeared [18]. The ability to absorb NH_4_^+^ in the *AtAMT1.1* gene knock-out plants was 30% lower than that of the wild-type [19]. Similar results were found in rice. *OsAMT1.1* was expressed in both rice roots and leaves under low N stress. However, the expression level of *OsAMT1.1* increased significantly in the roots when rice plantlets were transferred from the nutrient conditions of high N to low N [20]. Knockout of *OsAMT1.1* significantly decreased the total N transport from roots to shoots under low N stress, which suggested that the gene *OsAMT1.1* plays an important role in the N-K homeostasis in rice [16]. Some scholars obtained the *OsAMT1.1* overexpressed rice and observed an increase in the NH_4_^+^ absorption capacity and the NH_4_^+^ content in these plants [21,22], and the yield of the transgenic rice was about 30% higher than that of the wild-type under low N stress [9].

Due to its vital role in increasing NH_4_^+^ uptake, *AMT1.1* should have great potential in the development of varieties with high NUE in ammonium-preferring crops under low N stress. However, the physiological function of *AMT1.1* from sugarcane, as well as its mechanisms, remains unclear so far. Here, we identified an N deficiency-responsive *AMT1.1* gene based on the comparative transcriptional analysis between the low N-tolerant sugarcane cultivar ROC22 and the low N-sensitive sugarcane cultivar Badila. The *AMT1.1* gene, termed *ScAMT1.1*, was subsequently cloned from ROC22. Then, the *ScAMT1.1* gene was stably transformed into rice to explore the ammonium assimilation process of transgenic plants under low N stress. The results help us to preliminarily understand the biological function of sugarcane *ScAMT1.1* and lay a foundation for the molecular improvement of sugarcane with high NUE in the future.

## 2. Results

### 2.1. The ScAMT1.1 Gene Responded Positively to Low N Stress in Sugarcane

From samples of leaf and root in two sugarcane varieties ROC22 and Badila treated with low N stress, five differentially expressed genes (DEGs) of the AMT family were identified based on the previous mRNA transcriptomic datasets generated by RNA-Seq [23]. Their expression patterns in sugarcane under low N stress were evaluated by both RNA-Seq and quantitative real-time PCR (qRT-PCR) using the samples of roots and leaves, indicating the consistency of their expression patterns in both detections (Figure 1). In detail, *ScAMT1.2* or *ScAMT2.1* was upregulated in the roots of both ROC22 and Badila, as well as in the leaves (Figure 1B,C). Contrary to *ScAMT1.2* and *ScAMT2.1*, *ScAMT3.2* presented a downregulated pattern in two tissues of both high- and low-NUE sugarcane cultivars (Figure 1D). In addition, *ScAMT3.3* was specifically downregulated in the roots (Figure 1E). Moreover, the expression levels of *ScAMT1.2*, *ScAMT2.1*, *ScAMT3.2*, and *ScAMT3.3* were not significantly different between the two cultivars under low N conditions. Interestingly, *ScAMT1.1′* expression profiles were different from the other four *ScAMTs*. Within those AMT family members, only *ScAMT1.1* was kept at higher levels in ROC22 than that in Badila under low N concentration (Figure 1A). The higher expression level of *ScAMT1.1* might contribute to improving the low N tolerance and high NUE ability of ROC22.

### 2.2. Gene Cloning and Bioinformatics Analysis of ScAMT1.1 Gene

In the present study, a *ScAMT1.1* gene was cloned from the sugarcane cultivar ROC22 (Appendix A). Its full-length cDNA was 1868 bp, containing a 1491 bp open reading frame (ORF) and encoding 496 amino acids (Appendix A). A conservative AMT1.1 domain, belonging to the AMT superfamily, was located at the middle region (from 36th to 459th amino acid residues) of the ScAMT1.1 protein sequence (Figure 2). The theoretical molecular weight of the ScAMT1.1 protein was 124.47 kDa, which belongs to the acidic hydrophobic protein and contains transmembrane domains with the greatest probability of localization in the plasma membrane.

The ScAMT1.1 from sugarcane and the AMT family from the other four species, i.e., *A. thaliana*, *O. sativa*, *S. bicolor*, and *S. spontaneum*, were used to generate a phylogenetic tree. The results showed that the genetic relationship coincided with the plant AMT class (Figure 3). ScAMT1.1 was classed into the AMT1.1 group and had the closest genetic relationship with SbAMT1.1 (Figure 3). The multiple sequence alignment result was similar to the phylogenetic analysis. The amino acid sequence of ScAMT1.1 shares 98.19%, 96.18%, 91.57%, 85.29%, 73.67%, and 72.87% identity with AMT1.1 from *S. bicolor*, *Z. mays*, *O. sativa*, *T. aestivum*, *G. max*, and *A. thaliana*, respectively (Figure 4). Sugarcane is one of the monocotyledonous C4 crops, alongside *S. bicolor* and *Z. mays*. Therefore, the phylogenetic analysis and sequence alignment results were consistent with their biological classification features (Appendix A).

### 2.3. Identification of the ScAMT1.1 Overexpressed Rice

The seeds from 20 transgenic rice lines were screened for response to herbicide treatment at germination, as well as the three-leaf stage. Among them, 19 lines were determined as the positive transgenic plants by seed germination and plant sprayed using Basta solution (Figure 5A,B) and detected using PCR (Figure 5C).

### 2.4. Phenotypic Index Analysis Shows Overexpressing ScAMT1.1 Enhances Plants’ Height and Fresh Weight in the Transgenic Rice under Low N Stress

Four homozygous T3 generation transgenic lines (T-2, T-5, T-6, and T-12) were obtained according to the methods described in 4.4. Compared to the other transgenic lines, higher *ScAMT1.1* expression levels in the transgenic lines of T-5, T-6, and T-12 were found by qRT-PCR (Figure 6A), and thus they were used to investigate the agronomic traits in the following assay. The transgenic lines showed a better performance than the wild-type plants under low N stress. After culturing for 16 d under low N nutrient solution, transgenic rice seedlings of T-5, T-6, and T-12 grew significantly better than the wild-type (Figure 6B), and their plant height was 46.83%, 26.58%, and 36.03% higher than that of the wild-type (Figure 6C), respectively. In addition, their fresh weight was 59.30%, 43.02%, and 52.33% higher than that of the wild-type, respectively (Figure 6D).

### 2.5. The Four Ammonium Assimilation Related Enzymes Were More Active in the Transgenic Lines Than That in the Wild-Type Plants

Co-expression analysis results showed that the expression levels of all the ammonium assimilation-related genes, except for the NADH-dependent glutamate synthase gene (*NADH*-*GOGAT1*), were significantly upregulated in the transgenic lines under low N condition (Figure 7A). Similar results were obtained from the experiments on enzyme activities (Figure 7B). No significant differences were found in the activity of the glutamate synthase (GOGAT) enzyme between the transgenic lines and the wild-type plants (Figure 7B). However, the activity of the other enzymes (glutamine synthetase, GS, and glutamate dehydrogenase, GDH) in the transgenic lines was significantly higher than those in the wild-type (Figure 7B).

### 2.6. Overexpressed ScAMT1.1 Enhanced Grain Number and Grain Yield of the Transgenic Plants in the Pot Experiment under Low N Condition

Under low N conditions, transgenic lines of the *ScAMT1.1* overexpressed rice and the wild-type were planted in the planting pots. The ripening time of transgenic lines was extended by about 15 d compared to the wild-type. However, the transgenic lines grew better than the wild-type during the whole growth duration of the rice. Figure 7A showed the different phenotypes of the wild-type and the *ScAMT1.1* transgenic rice at the grain filling stage. At harvesting time, the grain number per plant of the transgenic lines T-5, T-6, and T-12 was 6.87%, 4.72%, and 7.72% higher than that of the wild-type (Figure 8B), and the grain yield per plant was 11.80%, 7.64%, and 9.12% higher than that of the wild-type (Figure 8C), respectively. In the pot experiment, the *ScAMT1.1* overexpressed transgenic rice is more productive than the wild-type under low N conditions because its grain number and grain yield were 4.72–7.72% and 7.64–11.80% higher than that of the wild-type, respectively (Figure 8B,C).

## 3. Discussion

AMT is the most important ammonium transporter gene family in plants [24] and plays an important role in the absorption and transport of ammonium under low N stress [1]. The first *AMT* gene was isolated from *S. cerevisiae* [25]. So far, the AMT gene family has been identified and classified into four clusters (AMT1–AMT4) from various plant species, such as *O. sativa* [26] and *S. bicolor* [13]. Our previous study found five AMT-type DEGs (*AMT1.1*, *AMT1.2*, *AMT2.1*, *AMT3.2*, and *AMT3.3*), belonging to three AMT subfamilies (AMT1, AMT2, and AMT3), based on the comparative transcriptomic analyses of low N-response in two sugarcane cultivars with contrasting NUE [23]. Recently, genome-wide identification results showed that sugarcane wild species, *S. spontaneum*, contained six AMTs (*AMT2.1*, *AMT2.2*, *AMT3.1*, *AMT3.2*, *AMT3.3*, and *AMT4*) belonging to three AMT subfamilies (AMT2, AMT3, and AMT4) [7]. Surprisingly, according to this report, *AMT1.1* was absent in the AMT family in the sugarcane wild species *S. spontaneum*.

Within the AMT family, *AMT1.1* was an important member of the high-affinity transport system [1,9,16]. As a type of low ammonia concentration stress-sensitive gene, *AMT1.1* was found to express preponderantly in the roots in both *A. thaliana* [18] and *O. sativa* [20] under low N stress. Sugarcane is one of the few ammonium-preferring crops. In this study, one *AMT1.1* gene, termed *ScAMT1.1*, was cloned from the low N-tolerant sugarcane cultivar ROC22. Similar to the *AMT1.1* from *A. thaliana* or *O. sativa*, *ScAMT1.1* was induced under low N stress and highly expressed in the roots of sugarcane (Figure 1).

The other four members of the AMT family studied here exhibited a differential expression pattern in sugarcane under low N stress. *ScAMT2.1* was significantly upregulated in the leaves of sugarcane under low N treatment (0.6 mM N) for six hours (Figure 1). However, *ScAMT3.2* was significantly downregulated in the leaves at the same condition (Figure 1). The opposite expression trends were also found between *SsAMT2.1* and *SsAMT3.2* in *S. spontaneum* under low N stress (0.1 Mm N) [7]. Moreover, we found that the expression trends of *ScAMT2.1* and *ScAMT3.2* were similar under low N stress between two contrasting NUE sugarcane cultivars (Figure 1). However, it is worth noting that the expression level of *ScAMT1.1* was significantly higher in the low N-tolerant cultivar ROC22 than that in the low N-sensitive cultivar Badila (Figure 1). It could be speculated that *ScAMT1.1* played a positive role in the N absorption process in sugarcane.

N is typically supplied by a mixture of NH_4_^+^ and NO_3_^−^ in natural environments [2]. Most crops strongly preferred NO_3_^−^ [7]. While some crops, e.g., rice and sugarcane, showed a stronger preference towards NH_4_^+^ [7,8,9]. However, the reasons for the preference for NH_4_^+^ or NO_3_^−^ remain unclear [27]. AMTs are most likely involved in ammonium transport in a crop, which might be partially associated with the answer [28]. Therefore, the functions and molecular regulatory mechanisms of AMTs in NH_4_^+^ uptake cause particular attention. *AMT1.1* is an important ammonium transporter gene under low N stress. Whether overexpression of *AMT1.1* in rice can lead to an increase in the NH_4_^+^ absorption and yield production remained controversial.

Kumar et al. [21] and Huang et al. [22] overexpressed *OsAMT1.1* in rice. The NH_4_^+^ concentration was too high for transgenic plants to assimilate in time, thus resulting in toxicity and damage to plant growth. However, the results in another report were different [9]. The transgenic rice overexpressing *OsAMT1.1* showed significant enrichment of *OsAMT1.1* transcript under the nutrient solution of 30 μM (NH_4_)_2_SO_4_, which was 20 times higher than the wild-type [9]. At the same time, the expression levels of ammonium assimilation key genes (*GS1.1*, *GS1.2*, *GS2.1*, *Fd-GOGAT*, *NADH-GOGAT*, and *GDH*) in the transgenic plants were significantly increased [9]. After ammonium is absorbed into the roots, it is converted into glutamine (Gln) through GS. Gln and α-Ketoglutaric acid generate glutamic acid by the GS/GOGAT cycle, and inorganic N is converted into organic N. *GS1.2* and *NADH-GOGAT* were the key genes in the NH_4_^+^ assimilation of plant roots, and *GS1.1* and *Fd-GOGAT* were the key genes for N utilization in plants [9,29]. Therefore, the yield of the transgenic rice was about 30% higher than that of the wild-type [9].

To identify the function of *ScAMT1.1* under N-deficient conditions, homozygous transgenic rice lines overexpressed *ScAMT1.1* were developed in the present study. Our results were similar to Ranathunge et al. [9]. Heterologously expressed *ScAMT1.1* in rice enhanced the ammonium assimilation ability of the transgenic plants. The evidence was as follows. Firstly, the activity of ammonium assimilation key enzymes GS, and GDH, as well as the expression levels of ammonium assimilation-related genes (*GS1.1*, *GS1.2*, *GDH*, *Fd-GOGAT*, and *NADH-GOGAT2*), were all significantly higher in the transgenic plants than those in the wild-type (Figure 7). Secondly, the plant height and fresh weight of the transgenic rice seedlings were 36.48% and 51.55% higher than those of the wild-type under N-deficient conditions (Figure 6). Thirdly, grain number and grain yield per plant of the transgenic rice were 6.44% and 9.52% higher than those of the wild- type, respectively (Figure 8). These demonstrated that the ammonium assimilation ability of the transgenic rice was higher than that of the wild-type. It could be speculated that enhancement of NH_4_^+^ uptake in *ScAMT1.1*-overexpressed rice plants promotes the activity of enzymes involved in ammonium assimilation under low ammonium conditions. The NH_4_^+^ content in the wild-type and the transgenic rice, together with its absorption and transport mechanism in the transgenic rice, need to be determined in the future. Additionally, the potential for improved grain yield in the *ScAMT1.1* transgenic rice under low N stress also needs to be further detected in the field.

## 4. Materials and Methods

### 4.1. Materials and Treatments

Sugarcane leaf and root samples obtained from low N treatment and used in qRT-PCR were the same as those used in the previous study of RNA-Seq [23]. The leaves (low N treatment for 0 h, 6 h) and roots (low N treatment for 0 h, 3 h) of sugarcane cultivar ROC22 (low N-tolerant) and Badila (low N-sensitive) plantlets under low N condition (0.6 mM N) were selected. The samples of the leaves and roots from the cultivars ROC22 and Badila under low N treatment for 0 h were selected as the control, respectively.

The culture experiment of the transgenic rice under low N stress: seeds of the transgenic and wild-type rice were germinated and cultivated in the seedling trays. The plantlets at the three-leaf stage with uniform size were selected as experimental materials and divided into two groups as follows:

Group one, plantlets were cultured for 16 days in the low N nutrient solution (0.01 mM NH_4_NO_3_, 2.0 mM KCl, 1.0 mM KH_2_PO_4_, 2.5 mM CaCl_2_•2H_2_O, 0.5 mM Na_2_SiO_3_•9H_2_O, 0.025 mM Fe-EDTA, 1.0 mM MgSO_4_•7H_2_O, 20.0 μM H_3_BO_3_, 5.0 μM MnCl_2_•4H_2_O, 0.4 μM ZnSO_4_•7H_2_O, 0.2 μM CuSO_4_•5H_2_O, and 0.05 μM Na_2_MoO_4_•2H_2_O, pH 5.8), and the whole plant samples were collected. Ten randomly selected plantlets were pooled into a biological duplicate. Among them, six seedlings were used for physiological index determination, and the other four seedlings were frozen rapidly in liquid N for qRT-PCR of ammonium assimilation key genes. The experiments were done in three biological replicates. Total RNA of the transgenic and wild-type rice plantlets was extracted with TRIzol reagent (Invitrogen, Waltham, MA, USA) for the qRT-PCR experiment. RNA and cDNA were obtained referring to the method of the previous study [23].

The plantlets of the other group were cultivated in the planting pots (one seedling per pot). The pot sizes were 0.40 m in both diameter and depth, with eight 1.0 cm diameter holes located at the bottom. Each pot was filled with 10 kg of low N soil (pH 5.0 to 5.5) with an organic matter content of 7.09 g/kg, alkali-hydrolyzable N of 0.04 g/kg, effective phosphorus of 0.02 g/kg, rapidly available potassium of 0.10 g/kg, total N of 0.38 g/kg, total phosphorus of 0.28 g/kg, and total potassium of 20.1 g/kg. All planting pots were provided with the same amount of water and under the same cultivation management. Grain number and grain yield per plant were recorded at harvest time.

### 4.2. RNA-Seq Data Analysis and qRT-PCR Experiment

Based on RNA-Seq in the previous study [23], five differentially expressed (padj < 0.05 and a fold change ≥ 2) *AMT* family genes (*AMT1.1*, *AMT1.2*, *AMT2.1*, *AMT3.2*, and *AMT3.3*) were identified in the two contrasting NUE sugarcane cultivars (ROC22 and Badila) under low N condition. The expression profiles (in log_2_ Fold change) of the mentioned five genes in the leaves and roots were detected, respectively.

The qRT-PCR experiment was further used to examine the expression profiles of the five AMT family genes in the leaves and the roots of ROC22 and Badila under low N stress. In addition, the relative expression levels of ammonium assimilation key genes, e.g., *GS1.1*, *GS1.2*, *GDH*, ferredoxin-dependent glutamate synthase (*Fd-GOGAT*), *NADH-GOGAT1*, and *NADH-GOGAT2* were assessed in the transgenic rice. The glyceraldehyde-3-phosphate dehydrogenase (*GAPDH*) [30] and eukaryotic elongation factor1-alpha (*eEF-1a*) [31,32] were used as the reference genes in sugarcane and rice, respectively. The 2^−△△Ct^ method [33] was used to calculate the relative gene expression of the samples. The experiments were done in three biological replicates, and SYBR Green was used for training. The primer sequences of qRT-PCR in sugarcane and rice are shown in Appendix A, respectively.

### 4.3. Gene Cloning and Bioinformatics Analysis of Sugarcane ScAMT1.1 Gene

The samples of RNA and cDNA from the sugarcane cultivar ROC22 were obtained from the previous study [23]. A pair of cloning primers, *ScAMT1.1*-F (5’-GCCACACCCTCCCAATCC-3’) and *ScAMT1.1*-R (5′-ACACACTGAAAAAGGCAAGCAC-3’), was designed according to the unigene template from the transcriptome sequencing data. The PCR amplification procedure was consistent with the previous study [23]. The PCR product was ligated to pMD20-T cloning vector and transformed into *Escherichia coli* cells for proliferation. The plasmid was extracted and sequenced.

NCBI Conserved Domains (http://www.ncbi.nlm.nih.gov/Structure/cdd/wrpsb.cgi/ (accessed on 28 October 2022), ProtParam (https://web.expasy.org/protparam/ (accessed on 28 October 2022), ProtScale (http://web.expasy.org/protscale/ (accessed on 28 October 2022), TMHMM Server (http://www.cbs.dtu.dk/servers/TMHMM/ (accessed on 28 October 2022) and PSORT analysis (http://www.psort.org/ (accessed on 28 October 2022) were used to predict the conserved domain, physicochemical properties, hydrophobicity, transmembrane domain, and subcellular localization of ScAMT1.1 protein, respectively. Multiple sequence alignment was performed using DNAMAN 8.0 software. The phylogenetic tree of AMTs from various plant species was constructed by the neighbor-joining (NJ) method using MEGA 6.0. The chosen proteins were SsAMT2.1 (*S. spontaneum*/Sspon.03G0003380-1P), SsAMT2.2 (Sspon.03G0003380-1A), SsAMT3.1 (Sspon.03G0002070-1A), SsAMT3.2 (Sspon.01G0032110-2B), SsAMT3.3 (Sspon.04G0010470-1A), SsAMT4 (Sspon.01G0007620-1A). OsAMT1.1 (*O. sativa*/XP_015636241.1), OsAMT1.2 (XP_015623207.1), OsAMT1.3 (XP_015624850.1), OsAMT2.1 (XP_015639562.1), OsAMT2.2 (XP_015643018.1), OsAMT2.3 (XP_015621584.1), OsAMT3.1 (XP_015622013.1), OsAMT3.2 (XP_015630045.1), OsAMT3.3 (XP_015626434.1), OsAMT4.1 (Q10CV4.1). AtAMT1.1 (*A. thaliana*/NP_193087.1), AtAMT1.2 (NP_176658.1), AtAMT1.3 (NP_189073.1), AtAMT1.4 (NP_194599.1), AtAMT1.5 (NP_189072.1), AtAMT2 (NP_973634.1). SbAMT2.1 (*S. bicolor*/XP_002439939.1), SbAMT2.2 (XP_002458715.2), SbAMT3.1 (XP_002456706.1), SbAMT3.2 (XP_002466132.1), SbAMT3.3 (XP_002452249.1), SbAMT4.1 (XP_021307349.1), SbAMT1.1 (XP_021319114.1), and SbAMT1.2 (XP_002452468.1).

### 4.4. Generation of the Transgenic Rice Overexpressed ScAMT1.1

The ORF of *the ScAMT1.1* gene was subcloned into the *Bsa* I site of the binary plant expressed vector pBWA(V)BU, which was used to construct the overexpressed vector pBWA(V)BU-ScAMT1.1. The recombinant expression vector pBWA(V)BU-ScAMT1.1 was transformed into *Agrobacterium tumefaciens* strain EHA105 by electroporation. Then, the genetic transformation of rice (*O. sativa* L. sp. *Japonica* cv. Nipponbare) was carried out to obtain the T0 generation transgenic plants. These works were entrusted to BioRun Co., Ltd. (Wuhan, China).

From the generations T0 to T3, an assay for the transgenic rice with Basta resistance was as follows. Three hundred seeds of each transgenic line were taken and soaked in 0.1% Basta solution for germination. The rice seedlings germinated normally were planted in the seedling tray. When the seedlings were at the three-leaf stage, 0.2% Basta solution was used for the second spray screening. The plants with normal growth were transplanted into pots for soil culture to obtain seeds. The screening procedure was repeated. The seeds were firstly soaked with 0.1% Basta solution for screening, followed by spraying the seedlings with 0.2% Basta solution for further screening until the homozygous transgenic lines of the T3 generation were screened. The transgenic-positive plants were confirmed by PCR using *ScAMT1.1*F (5’-TGGGTTCATGCTCAAGTCCG-3’) and *ScAMT1.1*R (5’-ATAACAGGGTAATGCGGCCC-3’) primers.

### 4.5. Determination of Enzyme Activities Involved in Ammonium Assimilation

The activities of the ammonium assimilation key enzymes GS, GOGAT, and GDH were measured in the transgenic and wild-type rice plantlets under low N stress for 16 days. The GS activity was measured according to the method of Wang et al. [34]. The activities of GOGAT and GDH enzymes were determined according to the experimental methods of Groat and Vance [35].

## 5. Conclusions

We characterized a sugarcane ammonium transporter gene, *ScAMT1.1*, with a significantly higher expression level in the low N-tolerant cultivar ROC22 than that in the low N-sensitive cultivar Badila under low N stress. The activity of ammonium assimilation key enzymes (GS and GDH) and the gene expression levels of ammonium assimilation key genes (*GS1.1*, *GS1.2*, *GDH*, *Fd-GOGAT,* and *NADH-GOGAT2*) in the transgenic rice were all significantly higher than those in the wild-type under low ammonium condition, indicating that the *ScAMT1.1* has the ability to improve ammonium assimilation. Transgenic rice lines overexpressed *ScAMT1.1* show superior growth and significantly higher grain yield under low N stress in the pot experiment. Taken together, it is suggested that *ScAMT1.1* has the potential for improving ammonium assimilation, plant growth, and grain yield under low N fertilizer conditions; however, the potential for increasing production needs to be further verified in the field.

## Figures and Tables

**Figure 1 ijms-24-01595-f001:**
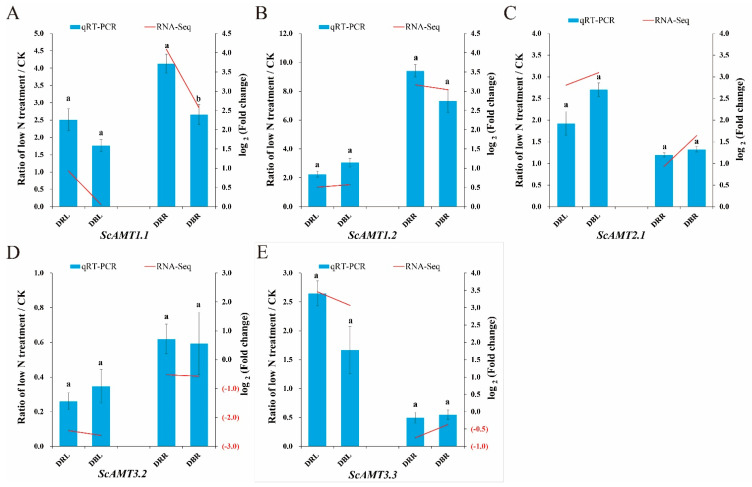
The expression patterns of the five AMT family DEGs between two sugarcane cultivars with contrastive NUE under low N condition. Notes: (**A**–**E**) represent the expression patterns of *ScAMT1.1*, *ScAMT1.2*, *ScAMT2.1*, *ScAMT3.2*, and *ScAMT3.3*, respectively. The expression patterns determined by qRT-PCR (the ratio of DEGs expression level between low N treatment and their control groups, Ratio of low N treatment/CK) and RNA-Seq (in log_2_ Fold change) are shown on the left and right sides in the same figure, respectively. DRL and DBL (RLT vs. RLCK, DRL. BLT vs. BLCK, DBL) mean the DEGs in the leaves of ROC22 and Badila under low N treatment (0.6 mM N) for 6 h vs. 0 h (used as control), respectively. DRR and DBR (RRT vs. RRCK, DRR. BRT vs. BRCK, DBR) mean the DEGs in the roots of ROC22 and Badila under low N treatment (0.6 mM N) for 3 h vs. 0 h (used as control), respectively. Error bars represent ± standard error from the mean of three biological replicates. Different letters above columns indicate significant differences (*p* < 0.05).

**Figure 2 ijms-24-01595-f002:**
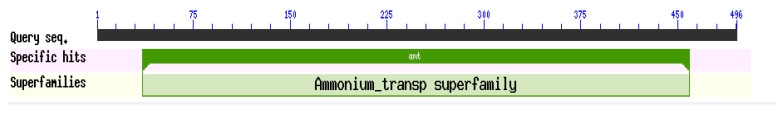
The conserved domain of the *ScAMT1.1* gene.

**Figure 3 ijms-24-01595-f003:**
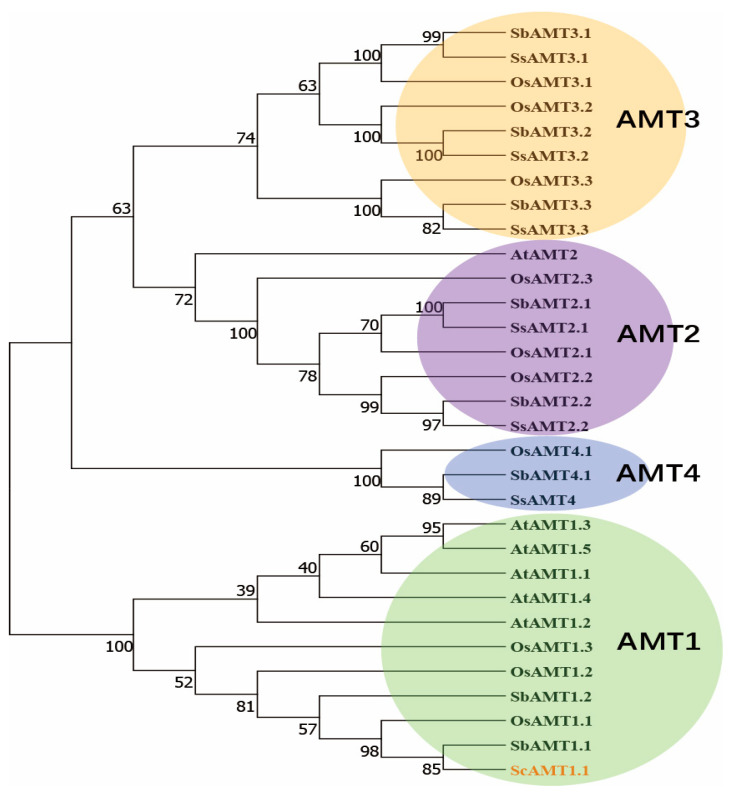
Phylogenetic relationship of sugarcane ScAMT1.1 protein and AMTs from *S. spontaneum*, *A. thaliana*, *O. sativa*, and *S. bicolor*.

**Figure 4 ijms-24-01595-f004:**
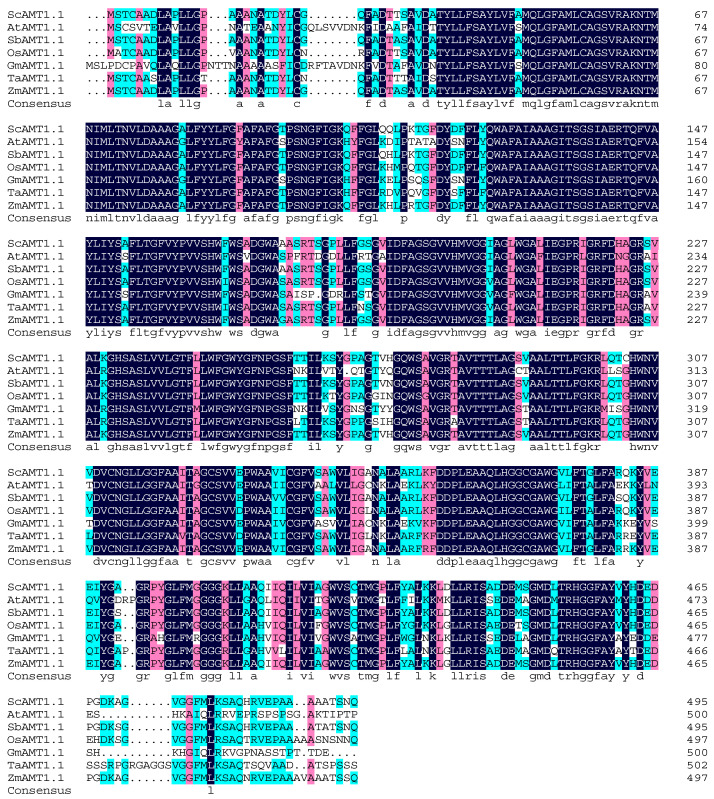
Sequence alignment of ScAMT1.1 and AMT1.1s from other species. Notes: *S. bicolor* (SbAMT1.1, AGK25010.1), *Z. mays* (ZmAMT1.1, NP_001140828.1), *O. sativa* (OsAMT1.1, Q7XQ12.1), *T.aestivum* (TaAMT1.1, AAS19466.2), *G. max* (GmAMT1.1, XP_003555734.1), and *A. thaliana* (AtAMT1.1, EFH51425.1).

**Figure 5 ijms-24-01595-f005:**
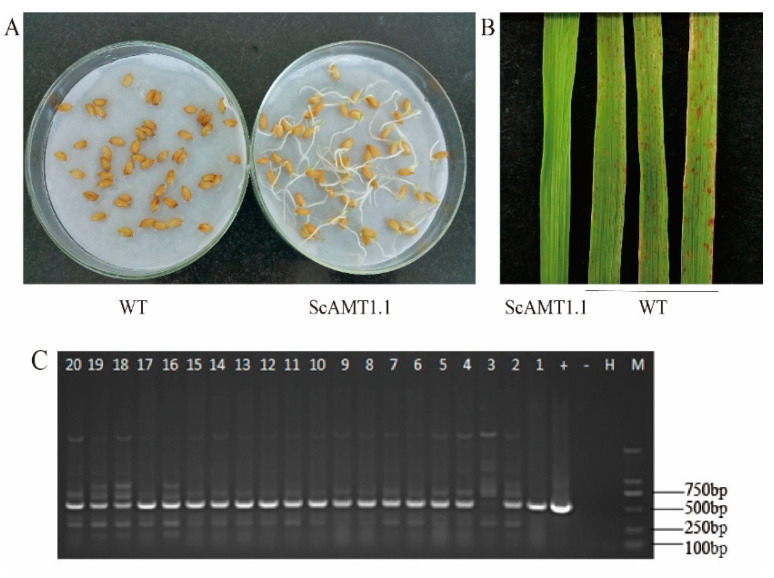
Identification of transgenic-positive rice plants. Notes: (**A**), The seeds germinate during soaking with 0.1% Basta solution. (**B**), The typical phenotype of the leaves spraying with 0.2% Basta solution. (**C**), Analysis of PCR amplification products by gel electrophoresis. WT: wild-type; ScAMT1.1: *ScAMT1.1*-overexpressing transgenic rice. M: 2000 bp DNA marker; H: H_2_O (blank control); -: negative control (wild-type rice); +: positive control (pBWA(V)BU-ScAMT1.1 plasmid), 1–20: different transgenic rice lines. The PCR product for the target sequence is 512 base pairs.

**Figure 6 ijms-24-01595-f006:**
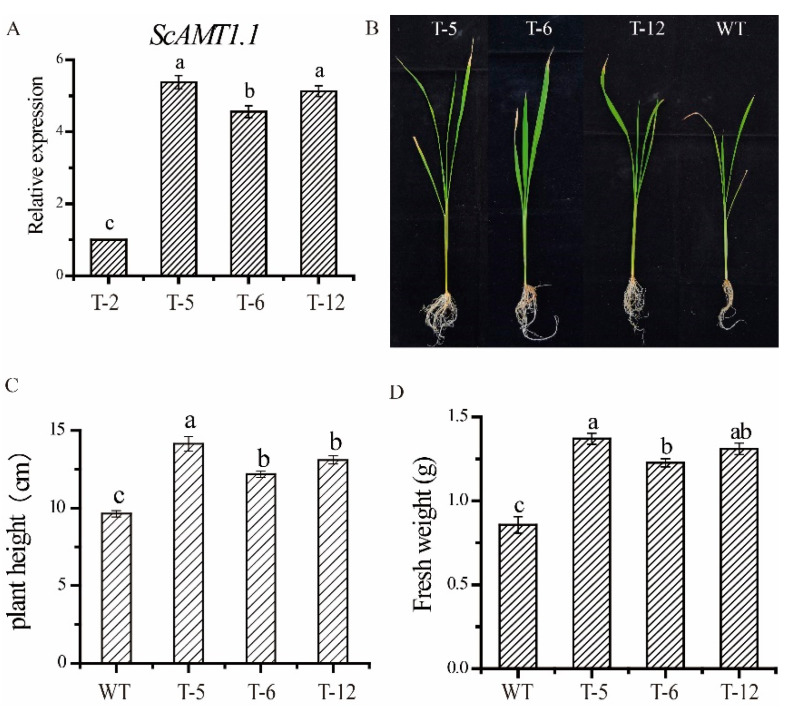
The transcription levels of *ScAMT1.1* and the morphological parameters of the transgenic rice seedlings under low N stress. Notes: (**A**), Transcription level detection of the *ScAMT1.1* gene in its transgenic rice plants. (**B**–**D**) represent the phenotype, plant height, and fresh weight of the transgenic and wild-type rice seedlings under low N stress, respectively. T-2, T-5, T-6, and T-12 represent the different transgenic lines. WT, wild-type. Data were mean ± standard error of three biological replicates. Bars with different letters were significantly different at the *p* < 0.05 level.

**Figure 7 ijms-24-01595-f007:**
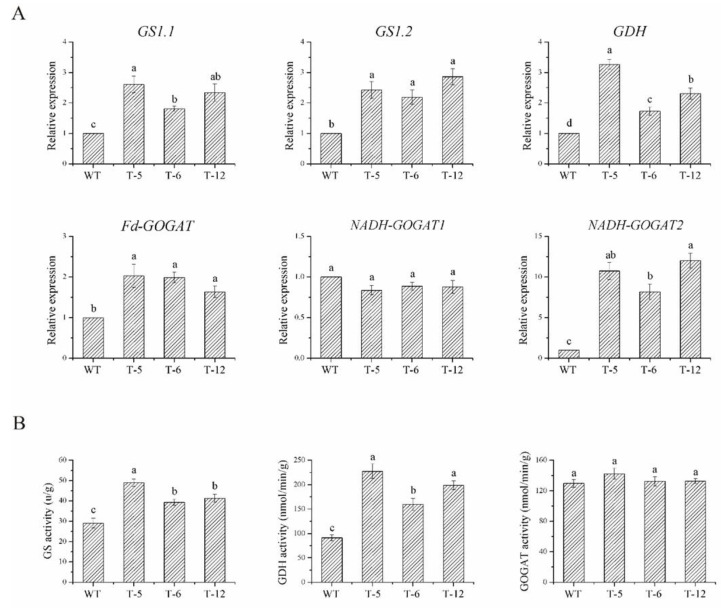
The gene expression levels and enzyme activities of ammonium assimilation-related enzymes in the transgenic rice under low N stress. Notes: (**A**), The expression levels of ammonium assimilation-related enzyme genes in the transgenic rice under low N stress. (**B**), The activities of three ammonium assimilation-related enzymes in the transgenic rice under low N stress. WT: wild-type. T-5, T-6, T-12: different transgenic lines. Data were presented as the mean ± standard error (*n* = 3). Bars with different letters were significantly different at the *p* < 0.05 level.

**Figure 8 ijms-24-01595-f008:**
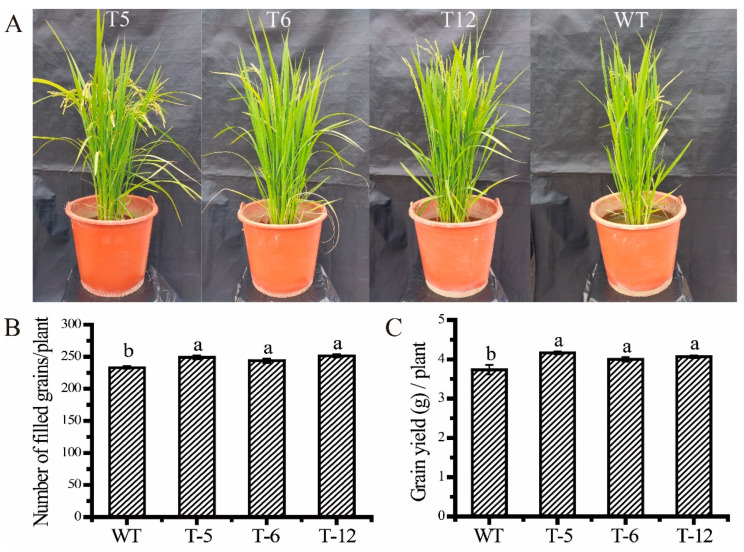
Phenotype and yield components of the transgenic rice in the pot experiment under low N stress. Notes: (**A**), The phenotype of the transgenic rice at the late filling stage under low N stress. (**B**,**C**), The number of filled grains and grain yield per plant of the transgenic rice under low N stress, respectively. WT: wild-type. T-5, T-6, and T-12: different transgenic lines. Data were presented as the mean ± standard error (*n* = 3). Bars with different letters were significantly different at the *p* < 0.05 level.

## Data Availability

Not applicable.

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
