# Peer review of "Ectopic Expression of Sugarcane ScAMT1.1 Has the Potential to Improve Ammonium Assimilation and Grain Yield in Transgenic Rice under Low Nitrogen Stress"

_ijms, 2023, doi:10.3390/ijms24021595_

Round 1
Reviewer 1 Report
Sugarcane is the most important sugar crop in China, which contributes more than 80% to the total sugar production in China. However nitrogen fertilizer is excessively used on sugarcane in China, while the nitrogen use efficiency (NUE) is relatively low. Identification of key gene of sugarcane in response to low N stress can provide target gene for developing sugarcane varieties with high NUE. In this work, ScAMT1.1 gene was found responding positively to low N stress and was cloned from sugarcane. Further over-expression of ScAMT1.1 gene in rice showed that the plant height and fresh weight of transgenic rice were significantly higher than that of the wild-type, respectively. Similar results also happened on the grain number and grain yield per plant in transgenic rice. These results indicate that sugarcane ScAMT1.1 gene is involved in ammonium assimilation under low N fertilizer conditions, which provides an important target gene for improving sugarcane varieties with high NUE by molecular technology. The evidences in this work are sufficient, the paper is well structured and written.
There is only one minor suggestion:
Line 126, ScAMT1.1 protein had a conservative AMT1.1 domain at the amino acids from positions ...”at the amino acids” may be “at N-terminus” or “at amino terminus”.
Reviewer 2 Report
In this article, the authors identified and cloned an ammonium transporter gene, ScAMT1.1, in sugarcane. Importantly, authors verified the biological function of this gene in low N adaptation by using transgenic rice. The topic is very interesting. However, some experimental designs are slightly inadequate. The detailed comments as follow:
(1) The description of lines 97-111 was inconsistent with Figure 1. Figure 1 only presented the expression difference of these genes between the two sugarcane cultivars and could not show the response of these to low N.
(2) For Figure 3, the conservative AMT1.1 domain should be presented.
(3) For Figure 4C, authors need to state what the “negative control” and “positive control” represent.
(4) Figure 5A, the ScAMT1.1 expression levels in the transgenic lines need to be compared with wild-type/non-transgenic rice.
(5) ScAMT1.1 was an ammonium transporter gene. Authors should determine the content of NH4+ in wild-type and transgenic rice to verify whether ScAMT1.1 has the ability to transport NH4+.
(6) Abbreviation should be used throughout the manuscript.
(7) Grammar checks should be carried out for proper English.
(8) Please check the structure of format presentation for “Int. J. Mol. Sci.”, especially the format of reference.
Reviewer 3 Report
This manuscript can only be submitted as a note because the phenotypic data are not enough to justify the author’s conclusions.
The title should be revised. I suggest the following title “Ectopic expression of sugarcane ScAMT1.1 may improve ammonium assimilation in transgenic rice under low nitrogen stress”
In the Abstract, please state clearly your objectives and the reasons for undertaking this research study.
One of the most important drawbacks of this manuscript is that there really are no phenotypic data to back up the author’s claims.
Authors have characterized few plants in the pots. This has nothing to do with characterizing the real genetic potential of transgenic and WT plants in the field. Growing few plants in the pots under selective conditions can not be called “Phenotyping” and can not be used for measuring grain yield.
Since there are no phenotypic data to justify the author’s conclusions, this paper can only be re-submitted (if revised accordingly) as a note with the statement that the authors will need to grow these plants in the field under an experimental design replicated in at least two locations or years before they can state that ScAMT1.1 can improve ammonium assimilation and grain yield.
Grain yield is a highly quantitative trait that interacts with the environment and is highly sensitive to environmental conditions. Measuring grain yield in pots, as part of phenotype characterization, is not rational for any practical plant breeding and genetics program.
Round 2
Reviewer 2 Report
No
Author Response
Thank you very much for your kindly agreeing to publish the paper.
Reviewer 3 Report
The authors have made some improvements but the major problem remains.
There really are no phenotypic data to justify the conclusions.
I am aware that this research is limited by the policies on genetically modified crops, and phenotyping under field-environmental conditions remains a bottleneck for many laboratories.
The fact that this research is limited by the existing policies does not mean that we can limit the quality of the phenotypic data and draw hypothetical conclusions based on pot experiments.
As I noted in my previous report, I believe that this manuscript can only be submitted and published as a note because the phenotypic data are not enough to justify an original paper.
Author Response
Thanks very much. During the revision, we all authors have tried our best to incorporate all your comments and suggestions, and this has no doubt greatly improved the quality of our manuscript. We believe that the data presented are sufficient to support our results and conclusions, and we therefore again strongly hope that the revised manuscript can be published as a paper rather than a note. Thanks again. What you have done will be highly and greatly appreciated.
Round 3
Reviewer 3 Report
The sentence in lines 309-312 does not read well towards the end. Please rephrase. I also noticed some other minor syntax errors throughout the text.
